# Inferring Causal Dependencies between Chaotic Dynamical Systems from Sporadic Time Series

**Edward De Brouwer** [* 1]  **Adam Arany** [* 1]  **Jaak Simm** [1]  **Yves Moreau** [1]

## Abstract

Discovering causal structures of processes is a major tool of scientific inquiry because it helps us better understand and explain the mechanisms driving a phenomenon of interest, thereby facilitating analysis, reasoning, and synthesis for such systems. However, accurately inferring causal structures within a phenomenon based on observational data only is still an open problem. In particular, this problem becomes increasingly difficult when it relies on data with missing values. In this article, we present a method to uncover causal relations between chaotic dynamical systems from *sporadic* time series (that is, incomplete observations at infrequent and irregular intervals), which builds upon Convergent Cross Mapping and recent advances in continuous time-series modeling (GRU-ODE-Bayes).

## 1. Introduction

Inferring a right causal model of a physical phenomenon is at the heart of scientific inquiry. It is fundamental to how we understand the world around us and to predict the impact of future interventions (Pearl, 2009). Correctly inferring causal pathways helps us reason about a physical system, anticipate its behavior in previously unseen conditions, design changes to achieve some objective, or synthesize new systems with desirable behaviors. As an example, in medicine, it allows to predict if a drug will be effective for a specific patient, or in climatology, to assess human activity as a causal factor in climate change. Causal mechanisms are best uncovered by making use of interventions because this framework leads to an intuitive and robust notion of causality. However, there is a significant need to identify causal dependencies when only observational data is available, because such data

is more readily available as it is more practical and less costly to collect (*e.g.*, relying on observational studies when interventional clinical trials are not yet available).

However, real-world data arising from less controlled environment than, for instance, clinical trials poses many challenges for analysis. Confounding and selection bias come into play, which bias standard statistical estimators. If no intervention is possible, some causal configurations cannot be identified. Importantly, with real-world data comes the major issue of missing values. In particular, when collecting longitudinal data, the resulting time series are often *sporadic*: sampling is *irregular* in time and incomplete across dimensions leading to varying time intervals between observations of a given variable and typically multiple missing observations at any given time. This problem is ubiquitous in various fields, such as healthcare (De Brouwer et al., 2019), climate science (Thomson, 1990), or astronomy (Cuevas-Tello et al., 2010).

A key problem in causal inference is to assess whether one time series is causing another or is merely correlated with it. From assessing causal pathways for neural activity (Roebroeck et al., 2005) to ecology (Sugihara et al., 2012) or healthcare, it is a necessary step to unravel underlying generating mechanisms. A common way to infer causal direction between two time series is to use Granger causality (Granger, 1969), which defines "predictive causality" in terms of the predictability of one time series from the other. A key requirement of Granger causality is then separability (i.e., that information about causes are not contained in the caused variable itself). This assumption holds in purely stochastic linear systems, but fails in more general cases (such as weakly coupled nonlinear dynamical systems) (Sugihara et al., 2012). To address this nonseparability issue, Sugihara *et al.* (Sugihara et al., 2012) introduced the Convergent Cross Mapping (CCM) method, which is based on the theory of chaotic dynamical systems, particularly on Takens' theorem. This method has been applied successfully in various applications in for example ecology, climatology (Wang et al., 2018), and neuroscience (Schiecke et al., 2015). However, as the method relies on embedding the time series under study with time lags, it is highly sensitive to missing values and, thus, cannot be applied in settings

---

*Equal contribution  [1]ESAT-STADIUS, KU Leuven, Belgium.  Correspondence to:  Edward De Brouwer <edward.debrouwer@esat.kuleuven.be>.

*Presented at the first Workshop on the Art of Learning with Missing Values (Artemiss) hosted by the $37^{th}$ International Conference on Machine Learning (ICML).* Copyright 2020 by the author(s).

with sporadic time series, despite their occurrence in many practical settings.

To address this important limitation, we leverage GRU-ODE-Bayes (De Brouwer et al., 2019), a recently introduced method that extends the Neural ODE (Chen et al., 2018) model. By relying on an Ordinary Differential Equation (ODE) model parameterized by a neural network to model the evolution of stochastic differential equations, it can handle sporadic time series. Based on sporadic observations, our approach learns the ODE dynamics to both probabilistically reconstruct the original data and then infer causal dependencies between the learned dynamical systems.

In a series of increasingly challenging test cases, our method accurately detects the correct causal dependencies with high confidence, even when fed very few observations, and outperforms other imputation methods, such as Gaussian Processes.

## 2. Related work

Granger causality (Granger, 1969) provides the first significant framework to infer causal dependencies from time series. Relying on predictability between dynamical systems, it was extended to account for different limitations, such as nonlinearity (Chen et al., 2004) or instantaneous relationships (Schiatti et al., 2015). However, the assumption of separability of information between causative and caused variables make the Granger paradigm fail for a significant number of time series coupling scenarios (Sugihara et al., 2012) (see Appendix D for an example). Convergent Cross Mapping, a technique based on nonlinear state space reconstruction was introduced to tackle this issue (Sugihara et al., 2012). Recently, Dimensional Causality (DC), an extension of CCM, was proposed to improve the discrimination of the confounding cases (Benkő et al., 2018).

Techniques to infer causal direction from non-fully-observed time series have also been proposed, all relying on the Granger causality framework. They use direct partial correlations on regularly sampled data (but with missing values) (Elsegai, 2019) or generalization of similarity measures for irregular time series (Bahadori & Liu, 2012). But, to the best of our knowledge, identifying causal dependencies from sporadic time series in the dynamical systems framework using CCM has not been investigated before.

## 3. Method

We consider here the problem of inferring causal dependency between two multivariate time series $X[t] \in \mathbb{R}^{d_X}$ and $Y[t] \in \mathbb{R}^{d_Y}$. Those time series are only observed at times $\mathbf{t}_X$ and $\mathbf{t}_Y$ respectively, which are typically not regularly spaced. As not all dimensions are sampled each time, we fur-

ther have two masks $M_X[t] \in \{0, 1\}^{d_X}$ and $M_Y[t]\{0, 1\}^{d_Y}$ for all observation times that indicate whether a given variable was observed at a given time. We say that $X$ causes $Y$ if $p(Y|do(X)) \neq P(Y)$ where $do(X)$ is an intervention on $X$ or, more loosely said, if intervening on process $X$ impacts the values of process $Y$.

### 3.1. Convergent Cross Mapping and Takens' theorem

CCM aims at discovering the causal direction between dynamical systems by checking if the value of one time series can be recovered from another. Intuitively, if $X$ causes $Y$, information from $X$ leaks into $Y$ and it should be possible to fully recover $X$ from $Y$, but not the other way round. The technique relies on Takens' embedding theorem (Takens, 1981). Let $X[t] \in \mathbb{R}^{d_X}$ be a chaotic dynamical system that has a strange attractor $A$ with box-counting dimension $d_A$. Takens' theorem states that a delay embedding with delay $\tau$

$$\phi(X[t]) = (\alpha(X[t]), \alpha(X[t-\tau]), \ldots, \alpha(X[t-k\tau]))$$

is an embedding of the strange attractor $A$ if $k > 2d_A$ and $\alpha : \mathbb{R}^{d_X} \to \mathbb{R}$ is a twice-differentiable observation function. More specifically, the embedding map $\phi$ is a diffeomorphism between the original strange attractor manifold $\mathcal{M}_\mathcal{X}$ and a shadow attractor manifold $\mathcal{M}'_\mathcal{X}$. Under these assumptions, one can then theoretically reconstruct the original time series from the delay embedding.

The simplest observation function $\alpha$ consists in simply taking one of the dimensions of the dynamical system. In this case, writing $X_i[t]$ as the $i$-th dimension of $X[t]$, Takens' theorem ensures that there is a diffeomorphism between the original attractor manifold of the full dynamical system and a shadow manifold $\mathcal{M}'_x$ that would be generated by $X'[t] = (X_i[t], X_i[t-\tau], \ldots, X_i[t-k\tau])$. To see how this theorem can be used to infer the causal direction, let us consider the manifold $\mathcal{M}_Z$ of the joint dynamical system resulting of the concatenation of $X[t]$ and $Y[t]$. We then generate two shadow manifolds $\mathcal{M}_X$ and $\mathcal{M}_Y$ from the delay embeddings $X'[t] = (X_i[t], X_i[t-\tau], \ldots, X_i[t-k\tau])$ and $Y'[t] : (Y_j[t], Y_j[t-\tau], \ldots, Y_j[t-k\tau])$. Now, if $X$ causes $Y$, because of Takens' theorem, it is theoretically possible to recover the original $\mathcal{M}_Z$ from $\mathcal{M}_Y$ and hence, by extension, recover $\mathcal{M}_X$ from $\mathcal{M}_Y$. However, the contrary is not true and it is in general not possible to recover $\mathcal{M}_Y$ from $\mathcal{M}_X$.

The CCM algorithm uses this property to infer causal dependency. It embeds both dynamical systems $X$ and $Y$ and use $k$-nearest neighbors to predict points on $\mathcal{M}_X$ from $\mathcal{M}_Y$ and inversely. The result then consists in the correlation of the predictions with the true values. If this correlation is high, we deduce that there is a causal arrow between the predictor dynamical system and the predicted one. Importantly, the

correlation will increase with the length of the observed time series, as the observed manifold becomes denser.

The potential results are then interpreted in the following way (1) $X$ causes $Y$ if one can reconstruct with high accuracy $\mathcal{M}_X$ from $\mathcal{M}_Y$; (2) $X$ and $Y$ are not causally related (but not necessarily statistically independent) if nor $\mathcal{M}_X$ nor $\mathcal{M}_Y$ can be reconstructed from the other; (3) $X$ and $Y$ are in a circular causal relation if both $\mathcal{M}_Y$ and $\mathcal{M}_X$ can be reconstructed from the other. In the extreme case of strong coupling, the two systems are said to be in synchrony.

### 3.2. Neural ODEs

Many continuous-time deterministic dynamical systems are usefully described as ODEs. One can thus attempt to describe the dynamics underlying a multivariate time series $X[t]$ as $\frac{dX(t)}{dt} = f_\theta(X(t), t)$ where we consider that the discretely sampled time series $X[t]$ comes from an underlying continuous process $X(t)$, and $f_\theta(\cdot)$ is a uniformly Lipschitz continuous function. Learning the dynamics of the system then consists in learning those parameters $\theta$ from a finite set of (potentially noisy) observations of the process $X$. Neural ODE (Chen et al., 2018) consists in parametrizing this function by a neural network, therefore allowing for a wide range of possible functions. Learning the weights of this network can be done using the adjoint method or by simply back-propagating through the numerical integrator.

### 3.3. Causal inference with GRU-ODE-Bayes

A key step in the CCM methodology is to compute the delay embedding of both time series: $X'[t]$ and $Y'[t]$. However, when the data is only sporadically observed at irregular intervals, the probability of observing the delayed samples $X_i[t], X_i[t-\tau], \ldots, X_i[t-k\tau]$ is vanishing for any $t$. $X'[t]$ and $Y'[t]$ are then never fully observed (in fact, only one dimension is observed) and nearest neighbor prediction cannot be performed.

As a first solution, casting the data into time bins would lead to more dimensions being observed. However, when the sampling is sparse, the binning would result in loss of accuracy without fully solving the problem because most samples would still contain missing values. This would result in very few samples in the state-space manifolds, therefore leading to low correlation scores in the CCM.

We propose to impute the sporadic time series by learning its governing ODE dynamics and then use those interpolated samples to compute the delay embeddings of both processes. In particular, we use GRU-ODE-Bayes (De Brouwer et al., 2019), a filtering technique that extends Neural ODEs. The method jointly learns the ODE driving the data and computes the filtered probability of future samples conditioned on previous ones, in continuous time. The filtering approach

is strictly causal, resulting in no leakage of future information backward in time. Crucially, this approach allows us to (1) compute the time delay embedding at all observed times $\mathbf{t}_X$ and $\mathbf{t}_Y$ and (2) reconstruct time delay embeddings at regular time points, using only the filter estimations, resulting is straightforward applicability of the CCM method for reliable causality direction estimation.

## 4. Experiments

We evaluate the performance of our approach on samples from the trajectories of three double pendulums. Based on their sporadic observations of the processes, we first independently compute a filtered continuous-time reconstruction for each dynamical system using GRU-ODE-Bayes and then use CCM to infer the causal dependencies.

### 4.1. Double pendulum

Each dynamical system in our experiments is a double pendulum, a simple physical system that is chaotic and exhibits rich dynamical behavior. It consists of two point masses $m_1$ and $m_2$ connected to a pivot point and to each other by weightless rods of length $l_1$ and $l_2$, as shown on Figure 1 in the appendix. The trajectories of the double pendulum are described by the time series of its state-space variables $\theta_1$ and $\theta_2$, defined as the angles of the rods with respect to the vertical, as well as the angular momenta $p_1$ and $p_2$ conjugate to these angles. Each trajectory is then a collection of 4-dimensional vector observations.

The time evolution of these pendulums is simulated by integrating the Hamiltonian using the Störmer-Verlet integrator. For the derivation of the Hamiltonian, see Appendix A.

To introduce causal dependencies, we include a non-physical asymmetrical coupling term in the update of the momentum conjugate to the first angle:

$$\dot{p}_1^X = -\frac{\partial H^X}{\partial \theta_1^X} - 2 \cdot c_{X,Y}(\theta_1^X - \theta_1^Y),$$

where $c_{X,Y}$ is a coupling parameter. The term corresponding to a quadratic potential incorporated to the Hamiltonian of system $X$ results in an attraction on system $X$ by system $Y$. Therefore, intervening on system $Y$ would result in change in system $X$, hence $Y$ causes $X$. Depending on the values of $c_{X,Y}$ and its reciprocal $c_{Y,X}$, we have different causal relationships between $X$ and $Y$. Namely,

- $X$ causes $Y$ iff $c_{Y,X} \neq 0$

- $Y$ causes $X$ iff $c_{X,Y} \neq 0$

- $X$ is not causally related to $Y$ if $c_{Y,X} = c_{X,Y} = 0$

*Table 1.* Average reconstruction correlations (and standard deviations) in all directions for Cases 1 (top) and 2 (bottom). Standard deviations are computed using 5 repetitions. Significant correlations are in bold. Our approach detects the correct causal structure. ✓ and ✗ highlight correct and wrong direction detection respectively.

| CASE | DIRECTION | LINEAR | GP | MVGP | OURS |
|------|-----------|--------|-----|------|------|
| CASE 1 | $X \leftarrow Y$ | $0.001 \pm 0.006$ ✓ | $-0.003 \pm 0.005$ ✓ | $-0.014 \pm 0.05$ ✓ | $0.0017 \pm 0.005$ ✓ |
|        | $\mathbf{X \rightarrow Y}$ | $0.000 \pm 0.004$ ✗ | $0.003 \pm 0.005$ ✗ | $-0.002 \pm 0.037$ ✗ | $\mathbf{0.209^* \pm 0.037}$ ✓ |
| CASE 2 | $X \leftarrow Y$ | $-0.0005 \pm 0.005$ ✓ | $-0.001 \pm 0.008$ ✓ | $-0.009 \pm 0.25$ ✓ | $0.0001 \pm 0.007$ ✓ |
|        | $X \rightarrow Y$ | $0.001 \pm 0.005$ ✓ | $0.001 \pm 0.003$ ✓ | $-0.007 \pm 0.019$ ✓ | $-0.019 \pm 0.06$ ✓ |
|        | $X \rightarrow Z$ | $0.003 \pm 0.007$ ✓ | $-0.001 \pm 0.002$ ✓ | $0.001 \pm 0.087$ ✓ | $-0.003 \pm 0.003$ ✓ |
|        | $\mathbf{Z \rightarrow X}$ | $0.001 \pm 0.007$ ✗ | $\mathbf{0.082 \pm 0.002}$ ✓ | $-0.013 \pm 0.033$ ✗ | $\mathbf{0.698 \pm 0.299}$ ✓ |
|        | $Y \rightarrow Z$ | $0.002 \pm 0.006$ ✓ | $0.001 \pm 0.003$ ✓ | $0.003 \pm 0.015$ ✓ | $0.003 \pm 0.012$ ✓ |
|        | $\mathbf{Z \rightarrow Y}$ | $0.002 \pm 0.005$ ✗ | $0.003 \pm 0.003$ ✗ | $0.0034 \pm 0.091$ ✗ | $\mathbf{0.096 \pm 0.048}$ ✓ |

## 4.2. Evaluation

We consider two main cases. The first with only two double pendulums ($X[t]$ and $Y[t]$) with $X$ causing $Y$. In this case, we set $c_{X,Y} = 0$ and $c_{Y,X} = 0.3$. In the second case, we consider three double pendulums ($X[t]$, $Y[t]$ and $Z[t]$) and investigate the ability of our method to identify confounding. We set $c_{X,Y} = c_{Y,X} = 0$, $c_{X,Z} = 0.5$, and $c_{Y,Z} = 0.8$, corresponding to the case where $Z$ is a confounder for both $X$ and $Y$. Graphical representation of those cases is presented in Appendix C along with the parameters of the pendulums (lengths and masses), as well as a third experiment case consisting of strong coupling between two double pendulums. For each of those cases, we generate 5 trajectories with different initial conditions ($\theta_1 \sim \mathcal{N}(-1, 0.05)$ and $\theta_2 \sim \mathcal{N}(0.5, 0.05)$). Each trajectory consists of 2,000 windows of 10 seconds. To simulate sporadicity, we sample observation uniformly at random with an average rate of 4 samples per second. Furthermore, for each of those samples, we apply an observation mask that keeps each individual dimension with probability 0.3. This leads to a sporadic pattern with missing observed dimensions at each sample as shown in Figure 2 of the appendix.

We trained GRU-ODE-Bayes on those samples, leaving, for each trajectory, 20% of the windows for hyperparameter tuning. We then computed the filtered reconstruction on *all windows* as there is no means for information about the target task (causal direction inference) to leak from this data. We apply CCM with an embedding dimension of $k = 10$ and with a time delay $\tau = 0.4s$, as dictated by the mutual information profile of the time series.

For each causal direction, we report the empirical correlations between predicted and actual samples in the delay embedding manifold. For instance, for a direction $X \rightarrow Y$, we report the correlation between predictions of $\hat{\mathcal{M}}_X$ obtained from $\mathcal{M}_Y$ and the actual ones ($\mathcal{M}_X$). Importantly, a strong positive correlation suggests an accurate reconstruction and thus a causal link in the studied direction between

both variables (*e.g.*, $X \rightarrow Y$). By contrast, a weak correlation suggests no causal link in that direction.

## 4.3. Baseline methods

To the best of our knowledge, this is the first time CCM is applied to sporadic time series. Indeed, because of missing variables, many standard approaches are simply not applicable. We compared our approach to an interpolation of the sporadic time series using (1) linear interpolation and (2) using univariate and multivariate Gaussian Processes (GP and MVGP). For the Gaussian Process, we chose a mixture of RBF and identity kernel and learn the parameters from the data. To model multivariate GPs, we used co-regionalization (Bonilla et al., 2008). We then use the mean of the posterior process as the reconstruction fed to the CCM method.

## 4.4. Results

Results over 5 folds for the two first cases (two and three double pendulums) are presented in Table 1. Regarding the first case, our approach is the only one to recover the right causal direction from the sporadic data. The other baselines do not detect any significant correlation and thus no causal link between both double pendulums. The second part of Table 1 presents the results for the second case (three double pendulums) in each direction. Again, our approach is the only one that infers the correct causal structure between the sporadic time series. Importantly, it shows that we can detect confounding whether the confounders are observed or not. Regarding the competing methods, the GP detects a weak causal relation between $Z$ and $X$, but fails to detect the one from $Z$ to $Y$. The linear method does not detect any edge. In Appendix C, we present another setup with stronger coupling where we show that our method is also the most reliable to infer the correct causal graph.

## 5. Conclusion and future work

In this work, we propose a way to detect causal structure linking chaotic dynamical systems that are sporadically observed using a neural ordinary differential equations model (GRU-ODE-Bayes). We show that our method correctly detects the causal directions between time series in a low and irregular sampling regime, even in the case of hidden confounders. Despite the apparent limitation of our method on chaotic systems, CCM is broadly applicable in practice as many real dynamical systems are indeed chaotic or empirically allow Takens'-like embeddings. We leave the application to other real world data as future work.

## Acknowledgements

YM is funded by Research Council KU Leuven: C14/18/092 SymBioSys3; CELSA-HIDUCTION CELSA/17/032 Flemish Government:IWT: Exaptation, PhD grants FWO 06260 (Iterative and multi-level methods for Bayesian multirelational factorization with features). This research received funding from the Flemish Government under the "Onderzoeksprogramma Artificiële Intelligentie (AI) Vlaanderen" program. EU: "MELLODDY" This project has received funding from the Innovative Medicines Initiative 2 Joint Undertaking under grant agreement No 831472. This Joint Undertaking receives support from the European Union's Horizon 2020 research and innovation program and EFPIA. EDB is funded by a FWO-SB grant.

## A. Double pendulum

Figure 1 presents a graphical representation of a double pendulum with its two masses and two weightless rods. Figure 2 shows examples of trajectories generated by a double pendulum.

The double pendulum is a simple physically system that is chaotic and exhibits rich dynamical behavior. The Lagrangian of the double pendulum is

$$\mathcal{L} = \frac{1}{2}(m_1+m_2)l_1^2\dot{\theta}_1^2 + \frac{1}{2}m_2l_2^2\dot{\theta}_2^2 + m_2l_1l_2\dot{\theta}_1\dot{\theta}_2\cos(\theta_1-\theta_2) \quad (1)$$

The corresponding Hamiltonian can be derived using Legendre transform $H = \sum_i \dot{\theta}_i p_i - \mathcal{L}$.

The system evolution can be simulated by integrating the Hamilton equations:

$$\dot{\theta}_i = \frac{\partial H}{\partial p_i}$$
$$\dot{p}_i = -\frac{\partial H}{\partial \theta_i}$$

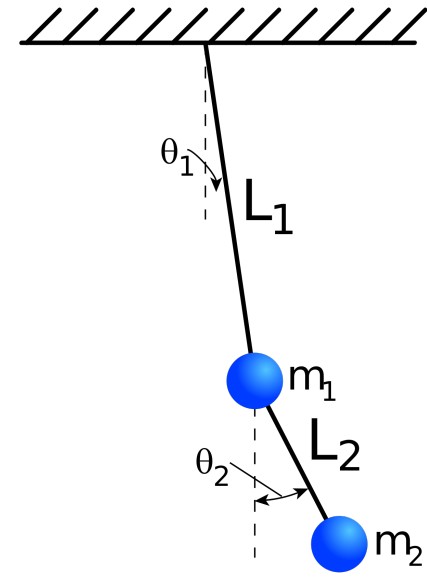

Figure 1. Physical representation of the double pendulum.

The Jacobian of the right hand side is

$$J = \begin{bmatrix} \frac{\partial^2 H}{\partial\theta_1\partial p_1} & \frac{\partial^2 H}{\partial^2 p_1} & \frac{\partial^2 H}{\partial p_1\partial\theta_2} & \frac{\partial^2 H}{\partial p_1\partial p_2} \\ -\frac{\partial^2 H}{\partial^2\theta_1} & -\frac{\partial^2 H}{\partial\theta_1\partial p_1} & -\frac{\partial^2 H}{\partial\theta_1\partial\theta_2} & -\frac{\partial^2 H}{\partial\theta_1\partial p_2} \\ \frac{\partial^2 H}{\partial\theta_1\partial p_2} & \frac{\partial^2 H}{\partial p_1\partial p_2} & \frac{\partial^2 H}{\partial\theta_2\partial p_2} & \frac{\partial^2 H}{\partial^2 p_2} \\ -\frac{\partial^2 H}{\partial\theta_1\partial\theta_2} & -\frac{\partial^2 H}{\partial p_1\partial\theta_2} & -\frac{\partial^2 H}{\partial^2\theta_2} & -\frac{\partial^2 H}{\partial\theta_2\partial p_2} \end{bmatrix}$$

Note the diagonal elements cancelling in pairs resulting in a trace of zero, indicating the volume preserving property of the Hamiltonian flow according to Liouville's theorem. This property corresponds to information preservation in nondissipating physical systems. Consequently, a noncoupled double pendulum does not have a proper attractor. However, for a given initial condition, and thus given energy, the possible states still form a densely populated volume in state-space. Applying the nonphysical coupling term, the conservation rule do not hold anymore.

The real part of the eigenvalues of $J$ are called the local Lyapunov exponents.

The direction of the largest expansion evolves as

$$\frac{d\mathbf{q}}{dt} = J\mathbf{q}$$
$$|\mathbf{q}(0)| = 1$$

The largest Lyapunov exponent is given by

$$\lambda_1 = \lim_{t\to\infty} \frac{1}{t}\log|\mathbf{q}(t)|.$$

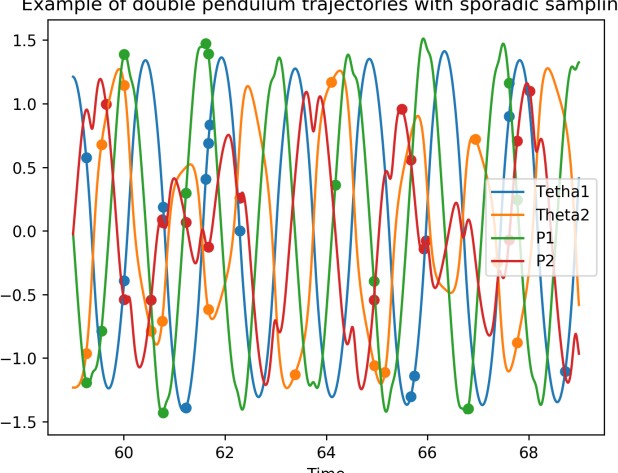

Example of double pendulum trajectories with sporadic sampling

*Figure 2.* Example of trajectories generated by a double pendulum. The solid lines represent the true process and the dots the sampled measurements.

Note that in stationary processes $J$ is constant, and the differential equation have a closed form solution

$$\mathbf{q}(t) = \mathbf{q}(0)e^{Jt},$$

and the local and global Lyapunov exponents are equal.

The largest Lyapunov exponent can be described intuitively as

$$|\delta(t)| \approx |\delta(0)|e^{\lambda_1 t},$$

where $\delta(t)$ is defined as the difference between two phase-space trajectories, with initial condition infinitesimally close to each other:

$$\mathbf{x}'(t) = \mathbf{x}(t) + \delta(t), t \geq 0$$
$$|\delta(0)| \leq \epsilon.$$

We use numerical integration to compute the largest Lyapunov exponent of the double pendulum, and verify it is in the chaotic regime.

## B. Case studies

Figure 3 shows a graphical model representation of the two main cases discussed in the body of the paper. The first one is a causal drive from $X$ to $Y$. The second case is with a third double pendulum $Z$ that acts as a confounder between $X$ and $Y$.

*Table 2.* Parameters [m, kg] and the largest Lyapunov exponents of the uncoupled pendulums ($\lambda_1 > 0$ indicates chaotic behavior). Numerical error given as standard deviation of 10 repetitions.

| SYSTEM | | $l_1$ | $l_2$ | $m_1$ | $m_2$ | $\lambda_1$ |
|---|---|---|---|---|---|---|
| $X \rightarrow Y$ | X | 1.0 | 0.5 | 2.0 | 1.0 | $0.398 \pm 0.0001$ |
| | Y | 0.5 | 1.0 | 0.5 | 4.0 | $0.005 \pm 0.0023$ |
| WHOLE $X \rightarrow Y$ SYSTEM | | | | | | $0.571 \pm 0.0014$ |
| $X \leftarrow Z \rightarrow Y$ | X | 0.5 | 1.0 | 2.0 | 1.0 | $0.008 \pm 0.0005$ |
| | Y | 1.0 | 0.5 | 0.5 | 4.0 | $0.505 \pm 0.0002$ |
| | Z | 1.0 | 1.0 | 1.0 | 3.0 | $0.010 \pm 0.0015$ |
| WHOLE $X \leftarrow Z \rightarrow Y$ SYSTEM | | | | | | $0.227 \pm 0.0009$ |

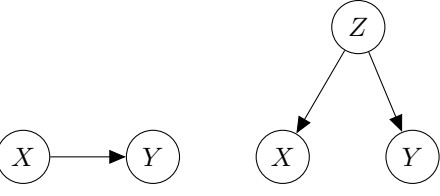

*Figure 3.* Graphical model representation of both cases considered in the main body of the paper. Left: Case 1. Right: Case 2 (confounding).

## C. Additional experiments

To further demonstrate the capabilities of our approach, we consider a third case involving two double pendulums with stronger coupling. It consists in 3 subcases. In Subcase 1, we set $c_{Y,X} = 1$ and $c_{X,Y} = 0$. In Subcase 2, $c_{Y,X} = 0$ and $c_{X,Y} = 1$. In Subcase 3, we consider no coupling. Graphical model representation of the 3 subcases are presented in Figure 4. The parameters (masses and lengths) of the double pendulums are the same as in Case 1 and are presented in Table 2 (top row). Again, we run our analysis over 5 repetitions.

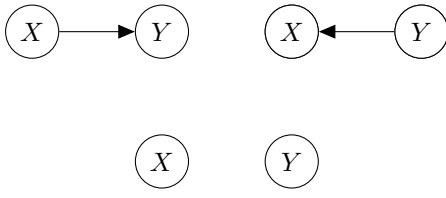

*Figure 4.* Graphical model representation of the three subcases. Top left: first subcase; $X$ drives $Y$. Top right: second subcase; $Y$ drives $X$. Bottom: third subcase; $X$ and $Y$ are not causally related.

Results for the five folds and 3 subcases are presented in Figure 5 and in Table 3. We report the causal direction score $\mathcal{S}_{X,Y}$, which we define as the difference in prediction accuracy between predicting $\mathcal{M}_X$ from $\mathcal{M}_Y$ and predicting

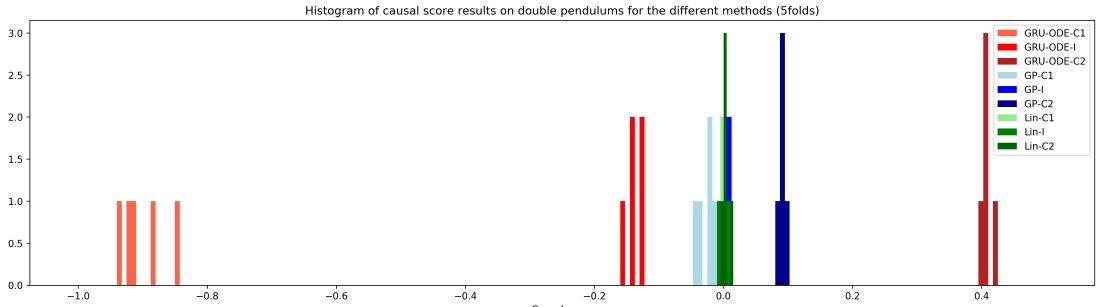

*Figure 5.* Distribution of the scores $\mathcal{S}_{X,Y}$ for the different scenarios (1, 2 and 3) and for the different methods. Our approach can infer causal direction reliably.

$\mathcal{M}_Y$ from $\mathcal{M}_X$. If we write $\hat{\mathcal{M}}_X$ as the reconstructed shadow manifold $\mathcal{M}_Y$ from $\mathcal{M}_Y$, the score is then defined as

$$\mathcal{S}_{X,Y} = \mathbb{Corr}(\mathcal{M}_X, \hat{\mathcal{M}}_X) - \mathbb{Corr}(\mathcal{M}_Y, \hat{\mathcal{M}}_Y), \quad (2)$$

where we abuse notation to indicate the correlation between the estimated values and the reconstructed ones on the shadow manifolds at each time $t$. Importantly, $\mathcal{S}_{X,Y}$ close to 1 suggests that $X$ causes $Y$, $\mathcal{S}_{X,Y}$ close to $-1$ suggests $Y$ causes $X$ and $\mathcal{S}_{X,Y} \approx 0$ suggests $X$ and $Y$ are not causally related or both cause each other. As we do not consider the latter case in our experiments, it would mean the former. Note that those cases could still be distinguished in practice by examining the individual value of both correlations in Equation 2

For Case 1 ($X \rightarrow Y$), our approach is close to optimal as it leads to a score of $-0.9$, close to $-1$. If it is able to detect some signal for Case 2 ($X \leftarrow Y$), it is however less confident about that dependency. Some spurious correlation is found for Case 3, leading to the score not being very close to 0. Overall, our method accurately detects the causal direction linking both time series. The baseline methods, by contrast, capture much less signal and all erroneously point towards no causal relationship between time series. Note that, while there is some consistency for the GP results (scores for Case 1 are lower than for Case 3, which are themselves lower than scores for Case 2), this does not allow to reliably infer causal dependencies, as one would conclude in all cases that no causal relations exist.

## D. Failure of Granger causality framework

To show how the Granger causality framework would fail in the general nonlinear dynamical systems case, we consider the following coupled dynamical system:

*Table 3.* Average scores (with standard deviations for all cases

| CASE | LINEAR | GP | OURS |
|---|---|---|---|
| $X \rightarrow Y$ | $0.001 \pm 0.002$ | $-0.027 \pm 0.01$ | $-0.9 \pm 0.03$ |
| $X \leftarrow Y$ | $0.004 \pm 0.004$ | $0.09 \pm 0.005$ | $0.4 \pm 0.01$ |
| $X \perp Y$ | $0.001 \pm 0.006$ | $0.003 \pm 0.004$ | $-0.13 \pm 0.01$ |

$$X[t+1] = X[t](a - bX[t] - cY[t])$$
$$Y[t+1] = Y[t](d - eY[t]).$$

Following Granger causality, including values of $Y$ for predicting $X[t+1]$ should increase the prediction accuracy, and thus hint towards a causal effect of $Y$ on $X$. However, dynamics of $X[t]$ can be rearranged such that all information about $Y[t]$ is contained in $X[t]$ already. Indeed,

$$Y[t] = \frac{-1}{c}\left(\frac{X[t]}{X[t-1]} - a + b\right)\left(d + \frac{e}{c}\left(\frac{X[t]}{X[t-1]} - a + b\right)\right).$$

Conditioning on $Y[t]$ would not bring additional information and Granger causality would then fail to uncover the right causal structure.

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
