# OpenReview forum: "Inferring Causal Dependencies between Chaotic Dynamical Systems from Sporadic Time Series"
_ICML.cc/2020/Workshop/Artemiss — ICML Artemiss 2020_

### Official Review · AnonReviewer1 · 2020-06-23
**Good paper, interesting and relevant topic, sound experiment.**

**Confidence:** 3
**Rating:** 7

**Review:**

Summary:
The paper proposes a new approach for inferring causal dependencies between sporadically observed, chaotic dynamical systems.
This is achieved via the combination of Convergent Cross Mapping (CCM) with GRU-ODE-Bayes, whereas GRU-ODE-Bayes is used to
learn an ODE dynamics which can be sampled arbitrarily to conform with CCM's restrictions of data availability at regular times.

Assessment:
The paper addresses an interesting and relevant problem: Causal inference on sporadic time series. To this end, the paper extends an existing method, CCM, to the sporadic time series setting via GRU-ODE-Bayes. In the experiments, for tackling the sporadic time series, linear interpolation as well as single- and multi-task GPs are compared against.
The paper shows strong results on tasks involving coupled double pendula and show that the proposed approach is uniquely able to infer the causal relationships correctly.

Strenghts:
- clear description of methods and setup
- strong empirical results in double pendulum setting

Weaknesses:
- Only 1 dataset / setting was studied, further datasets and application domains would strengthen the overall story.

Major points:
- 3.3 third paragraph: GRU-ODE-Bayes should be explained a bit more. E.g. what is meant with 'filtering technique', how does it extend Neural ODE? (here a brief supplemental section would be helpful to the reader)
- 4.2. penultimate paragraph: it would be helpful to explain in more detail how the CCM parameters were chosen using mutual information (e.g. suppl. section)
- 4.4 W.r.t. correlations, how is significance determined, i.e. which significance test?
- 4.4 I find it peculiar that the GP identifies the weak link between X and Z, but not the strong one between Y and Z. This should be investigated and discussed further.

Minor points:
- 4.1. last paragraph, penultimate line: 0(3) --> 0, and (3)
- 4.2. penultimate paragraph: mean for information.. --> means for information
- 4.4. last sentence: another setup we show --> another setup where we show

Comments:
- Following the last major point above, I would be very interested in a setup where both the dynamics and the CI is learnt end-to-end in a joint training process. However, for this I figure the CCM approach needed to be adapted. This came to my mind when seeing that the GP imputation showed strange effects, that weak links can be identified while stronger ones can be neglected.
In my own research, I have observed that the maximum likelihood solution of GP interpolations does not always correspond to the most useful interpolation depending on the specific downstream tasks (this motivated e.g. GP adapters).
- I wouldn't consider myself an expert on CI specifically, but I could imagine that there could be alternatives to CCM that could be compared to for this specific task. So I would recommend to consider this aspect when extending the experiments for a full paper submission.

---

### Decision · Program_Chairs · 2020-07-02

**Decision:**

Accept

**Comment:**

We are very happy to inform you that your paper has been accepted for the Artemiss workshop. We will contact you soon to inform you about the details concerning the format of your presentation at the workshop, and the camera-ready version deadline. Please take into account the referee's comments to write the camera-ready version.